# Enhanced DNA Repair Pathway is Associated with Cell Proliferation and Worse Survival in Hepatocellular Carcinoma (HCC)

**DOI:** 10.3390/cancers13020323

**Published:** 2021-01-17

**Authors:** Masanori Oshi, Tae Hee Kim, Yoshihisa Tokumaru, Li Yan, Ryusei Matsuyama, Itaru Endo, Leonid Cherkassky, Kazuaki Takabe

**Affiliations:** 1Department of Surgical Oncology, Roswell Park Comprehensive Cancer Center, Buffalo, NY 14263, USA; masa1101oshi@gmail.com (M.O.); taeheeki@buffalo.edu (T.H.K.); Yoshihisa.Tokumaru@roswellpark.org (Y.T.); Leonid.Cherkassky@RoswellPark.org (L.C.); 2Department of Gastroenterological Surgery, Yokohama City University Graduate School of Medicine, Yokohama 236-0004, Japan; ryusei@yokohama-cu.ac.jp (R.M.); endoit@yokohama-cu.ac.jp (I.E.); 3Department of Surgery, Jacobs School of Medicine and Biomedical Sciences, State University of New York, Buffalo, NY 14263, USA; 4Department of Surgical Oncology, Graduate School of Medicine, Gifu University, 1-1 Yanagido, Gifu 501-1194, Japan; 5Department of Biostatistics & Bioinformatics, Roswell Park Comprehensive Cancer Center, Buffalo, NY 14263, USA; li.yan@roswellpark.org; 6Division of Digestive and General Surgery, Niigata University Graduate School of Medical and Dental Sciences, Niigata 951-8520, Japan; 7Department of Breast Surgery, Fukushima Medical University School of Medicine, Fukushima 960-1295, Japan; 8Department of Breast Surgery and Oncology, Tokyo Medical University, Tokyo 160-8402, Japan

**Keywords:** HCC, GSVA, GSEA, tumor microenvironment, DNA repair, transcriptome

## Abstract

**Simple Summary:**

We studied the relationship between enhancement of DNA repair and cancer aggressiveness, tumor immune microenvironment, and patient survival in 749 hepatocellular carcinoma (HCC) patients from 5 cohorts using a DNA repair pathway score. We show that the DNA repair pathway was enhanced by the stepwise carcinogenic process of HCC, notably in grade 3 compared to grade 1 or 2 HCC. DNA repair high HCC was associated with worse survival, elevated intratumor heterogeneity, and mutation load, but not with the fraction of immune cell infiltration nor cytolytic activity. The expression of proliferation- and other cancer aggressiveness-related gene sets was also increased. Interestingly, these features were more pronounced in low-grade compared to high-grade HCC. In conclusion, the DNA repair score may be used to understand the role of DNA repair pathways in patient prognosis and treatment sensitivity and be used to improve patient outcome. To our knowledge, this is the first study using DNA repair pathway-related gene set expression data to examine and validate the clinical relevance of DNA repair pathway activity in HCC.

**Abstract:**

Hepatocellular carcinoma (HCC) is one of the most common malignancies and a leading cause of cancer-related deaths worldwide. In this study, a total of 749 HCC patients from 5 cohorts were studied to examine the relationships between enhancement of DNA repair and cancer aggressiveness, tumor immune microenvironment, and patient survival in HCC, utilizing a DNA repair pathway score. Our findings suggest that the DNA repair pathway was not only enhanced by the stepwise carcinogenic process of HCC, but also significantly enhanced in grade 3 HCC compared with grade 1 and 2 tumors. DNA repair high HCC was associated with worse survival, elevated intratumor heterogeneity, and mutation load, but not with the fraction of immune cell infiltration nor immune response. HCC tumors with a DNA repair high score enriched the cell proliferation- and other cancer aggressiveness-related gene sets. Interestingly, these features were more pronounced in grade 1 and 2 HCC compared to grade 3 HCC. To our knowledge, this is the first study to use DNA repair pathway-related gene set expression data to examine and validate the clinical relevance of DNA repair pathway activity in HCC. The DNA repair score may be used to better understand and predict prognosis in HCC.

## 1. Introduction

Liver cancer is the fourth most common cause of cancer-related deaths worldwide, and about 90% of primary liver cancers are hepatocellular carcinoma (HCC) [1]. Despite improvements in diagnosis and management of HCC, prognosis remains poor, with a 5-year survival rate less than 40% [2]. A prognostic biomarker based on cancer biology can tailor treatments according to patient prognosis. 

The DNA repair mechanisms activated following DNA damage are essential to suppress carcinogenesis. Carcinogens or metabolic processes can trigger genetic alteration, leading to genomic instability and malignant transformation. These DNA repair mechanisms counteract threats to genomic integrity [3]. Hence, deficiencies in DNA damage repair pathways promote carcinogenesis. Many DNA repair mechanisms regulate gene functions as transcriptional factors or cofactors. 

The role of DNA repair genes in cancer progression is somewhat controversial and context dependent compared to that of carcinogenesis. Upregulation of a DNA repair-related gene is associated with tumor metastasis in some tumors like melanoma, whereas the inverse has been observed in other cancers. The DNA repair pathway has been linked to heterogeneity and immunity as well. The role of DNA repair genes in HCC is actively being investigated, but contradictory reports on some DNA repair-related genes have complicated our overall understanding of DNA repair in HCC progression. Though it is important to study the function of each individual DNA repair-related gene, it is more important to understand the role of the entire DNA repair pathway in HCC progression. 

We previously reported the utility of scoring the expression profile of hundreds of genes to understand the relationship between cancer and pathways such as the cell cycle and angiogenesis. For example, the G2M checkpoint pathway score identified poor survival in pancreatic and breast cancer patients, and it showed potential as a predictive biomarker for chemotherapy in breast cancer [4,5]. In this study, we aim to investigate the clinical relevance of DNA repair pathway activity using the “Hallmark DNA repair” gene set of the MSigDB Hallmark gene set variation analysis (GSVA) as the DNA repair score in HCC cohorts. We hypothesize that the enhancement of the DNA repair pathway is associated with worse survival in HCC patients. To test our hypothesis, we analyzed a total of 749 HCC patients from The Cancer Genome Atlas (TCGA) Liver Hepatocellular Carcinoma (TCGA-LIHC; *n* = 358) [6], GSE6764 (*n* = 75) [7], GSE89377 (*n* = 107), GSE56545 (*n* = 42), and GSE76427 (*n* = 167) [8] cohorts to examine the role of DNA repair activity in clinical outcomes.

## 2. Results

### 2.1. DNA Repair Pathway Was Enhanced by the Stepwise Carcinogenic Process of Hepatocellular Carcinoma (HCC)

In order to quantify the activity of the DNA repair pathway, we defined the DNA repair pathway score based on the gene set variation analysis (GSVA) score of the Molecular Signatures Database Hallmark DNA repair gene set (Appendix A), using the methodology we previously reported [4,5,9,10,11]. The DNA repair score correlated with expression of BRCA1 and BRCA2, arguably the most clinically significant DNA repair genes associated with cancer prognosis and treatment sensitivity (Appendix A; Spearman’s rank correlation coefficient (r) = 0.576 (*p* < 0.01) and 0.471 (*p* < 0.01), respectively). We hypothesized that the DNA repair pathway is enhanced by the stepwise carcinogenic process of hepatocellular carcinoma (HCC) because cells with excessive DNA damage cannot survive. To test this hypothesis, we first compared the DNA repair score through the stepwise carcinogenic process of hepatocellular carcinoma (HCC) in normal liver tissue, dysplasia, cirrhosis, low- and high-grade chronic hepatitis, and early and advanced HCC in the GSE6764, GSE89377, and GSE56545 cohorts. DNA repair was significantly enhanced in early to advanced HCC compared to normal, dysplasia, cirrhosis, and very early HCC in the GSE6764 cohort (Figure 1A; *p* < 0.001). These results were replicated in the GSE89377 cohort where DNA repair was significantly enhanced in HCC compared with dysplasia, cirrhosis, and chronic hepatitis (Figure 1A; *p* < 0.001), and in the GSE56545 cohort, which showed significantly enhanced DNA repair in HCC compared with surrounding normal liver tissue (Figure 1B). DNA repair was also significantly enhanced in the higher pathological grade HCC compared with the low-grade tumor in the GSE89377 cohort (Figure 1A). This result was validated by the TCGA cohort (Figure 1C; *p* = 0.002), which demonstrated significant enhancement of DNA repair in tumors with advanced American Joint Committee on Cancer (AJCC) staging (Figure 1C, *p* = 0.001).

### 2.2. DNA Repair High HCC Was Associated with Significantly Worse Survival 

Since DNA repair was enhanced in advanced HCC, we expected DNA repair high HCC to be associated with a worse clinical outcome. We divided the cohort into low and high DNA repair score groups using the median value as the cut-off. We found that a high DNA repair score was significantly associated with worse disease-free survival (DFS), disease-specific survival (DSS), and overall survival (OS) in the TCGA cohort (Figure 2; *p* < 0.001, *p* = 0.002, and *p* < 0.001, respectively). Furthermore, we investigated the association of each gene in the DNA repair score with DSS. More than half of the 150 genes that constitute the score were significantly associated with a worse DSS (Appendix A). These results suggest that the DNA repair score reflected not only one, but multiple gene expressions that relate to survival outcome. 

### 2.3. High DNA Repair Score Was Associated with High Microsatellite Instability, Intratumor Heterogeneity, and Mutation Load, but Not with Immune Cell Infiltration or Immune Response

It has been modeled that tumors with a high mutation load generate neoantigens that prime an anti-tumor T cell immune attack [12], and abnormal DNA repair has been reported to be often mutagenic in cancer [13]. We investigated the relationship between the DNA repair score and microsatellite instability (MSI) as well as the intratumor heterogeneity, mutation-, and neoantigen-related scores, as calculated by Thorsson et al. [14]. We found that DNA repair high HCC was significantly associated with high MSI and high intratumor heterogeneity (Figure 3A; *p* < 0.001 and *p* = 0.009). Furthermore, DNA repair high HCC was significantly associated with the mutation related-score, including the fraction altered, silent and non-silent mutation load, and single-nucleotide variant (SNV) and indel neoantigens (Figure 3B; *p* < 0.001, *p* = 0.003, *p* = 0.002, *p* = 0.003, and *p* = 0.011, respectively). There was no significant decrease in infiltration of the immune cells estimated by xCell algorithm in DNA repair high HCC, except for regulatory T cells (Treg) and M2 macrophages, which are both pro-cancerous immune cells and expected to be increased (Figure 3C, *p* < 0.001 and *p* = 0.002, respectively). T helper type 1 (Th1) and T helper type 2 (Th2) cells were both increased in DNA repair high HCC (Figure 3C; both *p* < 0.001). There was no significant difference between the other anti-cancer immune cells, including CD4^+^ T cells, M1 macrophages, and dendritic cells, except for CD8^+^ T cells (Figure 3C). There was no significant difference in the cytolytic activity score (Figure 3D; *p* = 0.503), which suggests that there is no overall immune killing activity related to DNA repair. These results are consistent with the gene set enrichment analyses (GSEA) of the hallmark gene sets that showed that immune-related gene sets, IFN-γ response, and inflammatory response were not enriched by the DNA repair score (Figure 3E). These findings suggest that the DNA repair high HCC is associated with a high intratumor heterogeneity and high fraction of CD8^+^ T cells, Th1 and Th2 cells, and low fraction of regulatory T cells and M2 macrophages, but not with mutations, immune cell infiltrations, or cytolytic activity, which is perhaps the most direct measurement of anti-tumor T cell immunity and reflects the overall cancer immunity of the tumor immune microenvironment.

### 2.4. DNA Repair High HCC Enriched Cell Proliferation- and Cancer Aggressiveness-Related Gene Sets

Given the lack of a significant association of cancer immunity with DNA repair activity, it was of interest to identify the Hallmark gene sets that enrich the DNA repair high HCC using GSEA. DNA repair high HCC significantly enriched all cell proliferation-related gene sets, including E2F targets, G2M checkpoint, MYC targets v1, MYC targets v2, and Mitotic spindle (Figure 4A; normalized enrichment score (NES) and false discovery rate (FDR); NES = 1.83 and FDR < 0.01, NES = 1.71 and FDR = 0.03, NES = 1.57 and FDR = 0.07, NES = 2.02 and FDR < 0.01, NES = 1.99 and FDR < 0.01, respectively). Furthermore, DNA repair high HCC enriched the cancer aggressiveness-related gene sets, including the unfolded protein response, PI3K/AKT/MTOR signaling, Mtorc1, Notch signaling, and WNT-β catenin signaling pathways (Figure 4B; NES = 1.73 and FDR = 0.03, NES = 1.68 and FDR = 0.03, NES = 1.57 and FDR = 0.07, NES = 1.34 and FDR = 0.20, NES = 1.37 and FDR = 0.19, respectively). Since the mitogen-activated protein kinase (MAPK) pathway has been reported to be associated with DNA damage response, we also investigated the association between DNA repair and the MAPK signaling gene set of the KEGG gene sets with GSEA. DNA repair high HCC significantly enriched the MAPK signaling gene set (Appendix A). These results suggest that DNA repair high HCC is highly proliferative and associated with aggressive biology. 

### 2.5. High DNA Repair Score Was Significantly Associated with Microsatellite Instability, Intratumor Heterogeneity, and Mutation Load in Pathological Grade 1 and 2 HCC, but Not in Grade 3

For some biomarkers associated with higher grades and more advanced stages, association with patient prognosis, and therefore clinical significance, is most evident in early stage disease. For instance, lymphovascular invasion in breast cancer is associated with a higher-grade cell proliferation MKI67 expression, and an advanced stage; however, it is associated with survival only in early-stage disease [15]. To this end, we hypothesized that DNA repair has the most clinical impact in low-grade HCC than in high-grade HCC. To test this, we compared the MSI, intratumor heterogeneity, and mutation-related scores between the low and high DNA repair score groups in pathological grade 1 and 2 (G1/G2), and grade 3 (G3) HCC in the TCGA cohort. Although there were no significant differences in MSI, intratumor heterogeneity, or indel neoantigens in both groups, DNA repair high HCC was significantly associated with high levels of MSI and multiple mutation-related scores (silent mutation rate score—median = 0.56, interquartile range (IQR): 1.35–0.76 in the low score group; median = 0.66, IQR: 0.43–0.92 in the high score group; non-silent mutation rate score—median = 1.66, IQR: 1.17–2.61 in the low score group; median = 1.89, IQR: 1.54–2.84 in the high score group) in grade 1/2 HCC. However, there was no significant association in grade 3 HCC (Figure 5; silent mutation rate score—median 0.61, IQR: 0.41–0.82 in the low score group; median = 0.74, IQR: 0.50–0.99 in the high score group; non-silent mutation score—median = 1.83, IQR: 1.25–2.57 in the low score group; median = 2.17, IQR: 1.55–2.72 in the high score group).

### 2.6. The Enrichment of Cell Proliferation- and Cancer Aggressiveness-Related Gene Sets to DNA Repair High HCC Was More Pronounced in Grade 1/2 Than in Grade 3 HCC

Next, we performed GSEA of the Hallmark gene sets to DNA repair high HCC in grade 1/2 and grade 3 HCC. As expected, DNA repair high HCC significantly enriched the gene sets similar to Figure 4. Interestingly, the enrichment score, which is analogous to the degree of enrichment, was higher in almost all of the gene sets that were significantly enriched due to DNA repair high HCC in grade 1/2 compared with grade 3 (Figure 6). These results suggest that DNA repair high HCC is associated with aggressive cancer biology and particularly pronounced in grade 1/2 tumors compared with grade 3 HCC. 

### 2.7. DNA Repair High HCC Was Significantly Associated with Worse Survival in Early HCC Patients, but Not in Advanced HCC Patients 

Following our results, that DNA repair high HCC was strongly associated with intratumor heterogeneity and aggressive cancer biology in early grade rather than in advanced HCC, we investigated patient survival comparing the low and high DNA repair score groups in early and advanced HCC. We found that DNA repair high HCC was significantly associated with worse DSS in grade 1/2, but not in grade 3 groups in the TCGA cohort (Figure 7A; *p* = 0.006, 0.033, and 0.072, respectively). This result was validated with an independent GSE76427 cohort that found DNA repair high HCC significantly associated with worse progression-free survival (PFS) in the Barcelona Clinic Liver Cancer (BCLC) stage A group, but not in the BCLC stage B or stage C group (Figure 7B; *p* = 0.033, 0.471, and 0.222, respectively). These results suggest that DNA repair high HCC is clinically more relevant in early HCC than in advanced HCC.

## 3. Discussion

In this study, a total of 749 HCC patients from 5 cohorts were studied to determine the relationship between enhancement of DNA repair and cancer aggressiveness, tumor immune microenvironment, and patient survival in HCC using DNA repair pathway scores. Our findings suggest that the DNA repair pathway was enhanced by the stepwise carcinogenic process of HCC and significantly enhanced in higher grade HCC compared with low-grade tumors. DNA repair high HCC was associated with worse survival, elevated intratumor heterogeneity, and mutation load, but not with the fraction of immune cell infiltration nor immune response. HCC tumors with a DNA repair high score enriched cell proliferation- and other cancer aggressiveness-related gene sets. Interestingly, these features were more pronounced in grade 1 and 2 HCC compared to grade 3 HCC. DNA repair high HCC was associated with elevated mutation load, enriched the proliferation- and other cancer aggressiveness-related gene sets, and was associated with worse survival in low-grade but not in high-grade HCC. To our knowledge, this is the first study to use DNA repair pathway-related gene set expression data to examine and validate the clinical relevance of DNA repair pathway activity in HCC.

Quantification of the DNA repair activity in solid tumors is challenging due to technical difficulties. DNA repair capacity can be measured by a direct assay or indirectly by cultivation of peripheral blood lymphocytes, but these assays are cost and labor intensive to conduct in large cohorts [16]. There are several publications that utilize a specific gene expression level to dissect the role of DNA repair in cancer progression. For example, RuvB-like2 (*RUVBL2*), which is responsible for the detection and repair of DNA damage, promotes cell proliferation and is upregulated in HCC, and is associated with a poor prognosis [17]. APEX is a known regulator that showed a positive correlation with the DNA damage repair signaling pathway and is suggested to be a potential prognostic biomarker for HCC patients. The expression of some DNA repair-related genes has been reported to be associated with chemoresistance in HCC [18,19]; however, other studies have suggested the inverse [20,21]. DNA repair involves many molecules, and analysis of a single gene expression may not reflect the entire pathway, which results in such discrepancies. Recent technological advances in high-throughput sequencing has helped us to understand the dynamics of damage repair at the genome-wide level [22]. DNA damage maps are expected to be important tools in the sequencing arsenal to study mutagenesis, carcinogenesis, and responsiveness to DNA damaging drugs. However, sequencing DNA damage in a reliable and robust manner still requires significant work. 

In this study, we utilized the DNA repair pathway score as defined by the gene set variation analysis (GSVA) algorithm, with the transcriptomes of multiple large HCC patient cohorts, to quantify the enhancement of DNA repair in HCC. Interestingly, the Hallmark DNA repair gene set does not include known DNA repair genes such as *BRCA1* and *BRCA2*, arguably the most clinically relevant DNA repair genes associated with cancers. This is because the Hallmark gene sets were determined by computational biological methods that identified overlaps between the other gene sets in the Molecular Signatures Database (MSigDB) and retained the genes that display coordinated expression, as determined by Liberzon et al. in *Cell Systems* [23]; thus, gene sets do not necessarily contain genes that are commonly known to be involved in experimental studies. On the other hand, the Hallmark gene sets are widely accepted and recognized as representing a specific and well-defined biological status or process and displaying coherent expression. The aim of this paper was not to generate a novel gene set that reflects DNA repair, but to investigate the clinical relevance of the existing Hallmark DNA repair gene set. We have previously reported the association of several pathways with clinical outcomes in various cancers. The G2M checkpoint pathway score identified poor survival in pancreatic and breast cancer patients, and it demonstrated potential as a predictive biomarker for chemotherapy in breast cancer [4,5]. The E2F target pathway score also demonstrated potential as a predictive biomarker for neoadjuvant therapy in breast cancer [10]. The angiogenesis pathway score revealed the association of intratumoral angiogenesis with inflammation, immune reaction, and metastatic recurrence in breast cancer [11]. These findings support the idea that pathway scores may be a useful tool in understanding the role of the DNA repair pathway in cancer. In this study, the DNA repair score was associated with various clinical features and outcomes.

The enhanced DNA repair pathway was expected to decrease the mutation rates and chromosomal rearrangements of the oncogenes and/or tumor suppressor genes that suppress the aggressiveness of cancer cells [24,25,26]. However, recent findings revealed that DNA repair can paradoxically lead to deleterious effects on genome stability. Hyperactivated DNA repair in cancer cells may support tumor maintenance, as constantly proliferating cancer cells need to cope with large amounts of DNA damage induced by the high replication stress and increased levels of oxidative stress [27,28]. Sy et al. reported that genomic instability in HCC may be caused by erroneous DNA repair in a desperate attempt to mend DNA double-strand breaks for cell survival, and such preemptive measures inadvertently foster chromosome instability and complex genomic rearrangements [29]. Wang et al. reported that the homologous recombination and nonhomologous end-joining DNA repair pathways were up-regulated to maintain HCC growth in vitro [30]. This explains why in this study HCC had a higher DNA repair score compared to normal and benign tissue, and a high DNA repair score was associated with tumor aggressiveness and worse survival in HCC patients. This is also consistent with our results demonstrating that DNA repair high HCC is associated with cancer cell proliferation with increased replication stress. 

Therefore, it was of interest to determine whether high DNA repair activity was associated with worse survival or if low activity was associated with better survival. Hyperactive DNA repair has been reported to contribute to genomic instability in HCC, underscoring the importance of a balanced DNA repair pathway in maintaining genomic stability, especially in cancerous cells that experience overwhelming amounts of DNA damage [29]. Impaired DNA repair and/or genomic instability can lead to increased mutagenicity in cancer cells, which increases the tumor antigen load, resulting in high levels of immunogenicity [31]. This notion of enhanced immunogenicity due to DNA repair deficiencies in tumors is consistent with the mismatch repair (MMR) abnormalities seen in colorectal cancers [32]. Indeed, inactivation of MMR increases the mutational load, promotes continuous renewal of neoantigens, and triggers immune surveillance. This results in suppressed tumor growth in human colorectal cancers in mouse models. MMR-deficient tumors also exhibited a better response to the blockade of programmed cell death protein 1 (PD-1) compared to MMR-proficient tumors because of increased immunogenicity in several cancers, including colorectal cancer and melanoma. Our group previously reported that high expression of Polo-like kinase 1 (*PLK1*), which plays a pivotal role both in the p53-mediated regulation of DNA damage repair and in mitosis, was associated with homologous recombination deficiency as well as high levels of CD8^+^ T cells, M0 and M1 macrophages, low levels of M2 macrophages, and high immune cytolytic activity in breast cancer [32,33]. In this study, the DNA repair high score was significantly associated with a high fraction of CD8^+^ T cells, Th1, and Th2, and low fraction of regulatory T cells and M2 macrophages, as well as high homologous recombination defects and intratumor heterogeneity; but, the score was not associated with cytolytic activity nor enrichment of the immune-related gene sets. We expected low-score tumors to be associated with a better prognosis due to a high fraction of anti-cancer immune cells; however, our study did not show an association between DNA repair low score and high anti-cancer immune cell infiltration in the tumor immune microenvironment. Rather, high-score tumors strongly enriched the cancer aggressiveness-related gene sets, explaining why high score tumors were associated with a poor prognosis. Recently, Yang et al. reported that the tumor mutation burden and neoantigen load were not associated with patient survival in HCC patients with a TP53 mutation, and they suggested that the TP53 mutation may have a unique effect on T cell infiltration [34]. We speculate that the tumor immune microenvironment of HCC may not depend on the tumor mutational burden and neoantigen load, resulting in the lack of association between DNA repair and overall cancer immunity in our study.

The results of GSEA, which illustrated the association between the DNA repair and cell proliferation pathways, provided one reason as to why the high DNA repair score group was associated with worse survival. Interestingly, the relationship between the DNA repair score and clinical features were particularly related in grade 1 and 2 HCC, which is in agreement with the notion that DNA repair activity is critical in carcinogenesis, the early stage of cancer development. On the contrary, our group previously found that abundant pathways are involved in high-grade compared with low-grade breast cancer [35]; thus, it is possible that DNA damage and repair may occur too extensively in high grade HCC for repair activity to be of clinical relevance. Further, since it is known that the survival of high-grade HCC patients is so severe [36], there is no variability in outcomes. High-grade cancer is affected by many malignant factors and pathways that a single pathway appears to not have a clinical impact. In agreement, we found that the survival of DNA repair score high patients in grade 1 was as poor as those in grade 2 or 3, whereas DNA repair score low patients in higher grades did worse than those in grade 1. Given that our DNA repair pathway score predicted worse outcomes in low-grade HCC, we cannot help but speculate that the score may be a useful tool for critical patient selection for costly and aggressive treatments, such as liver transplantation. Our findings may provide a hint for future clinical trials in limiting patient selection to low-grade HCC when targeting the DNA repair pathway. Although basic research that discovers and proves novel mechanisms using in vivo and in vitro experimental settings is essential to advance science, no system perfectly reproduces human cancer and its tumor microenvironment. To this end, translation of the mechanisms found in purely experimental systems into human patients is essential to make clinical impact. The aim of the current study was not to report a new mechanism of the DNA repair pathway in HCC, but rather to highlight the ability of the DNA repair score to predict patient survival in early HCC. Further, the DNA repair score comprises 150 genes, allowing us to consolidate the complex system of cancer biology, improving the explanatory power of the prediction model as a single scoring system. This system is stronger than single gene expression, which often suffers from limited reproducibility and difficulty interpreting its biological meaning [37,38,39]. As shown in Appendix A, more than half of the genes that constitute the DNA repair score were significantly associated with a poor prognosis. For instance, genes such as *SAC3D1, NELFCD,* and *TAF9* (TATA-box binding protein associated factor 9) were implicated in the DNA repair pathway and are associated with survival in some cancers [40,41,42]. *SAC3D1* plays a role in mitotic progression, centrosome assembly, and spindle assembly [43], which supports its prognostic significance in HCC [44]. On the contrary, SAC3 mutants were shown to delay the G2/M phase [45]. The *NELF* complex fosters double-strand break-induced transcription silencing and promotes homology-directed repair [42]. *TAF9* is the general transcription coactivator that interacts with the majority of transcription factors, such as p53 and nuclear factor-kB, via the 9aa TAD [46,47]. Our results suggest that the high DNA repair score was associated with poor prognosis due to the combined effects of these genes. 

Finally, as we have demonstrated that the MAPK signaling pathway is associated with DNA repair high HCC (Appendix A) and that tyrosine kinase inhibition by Lenvatinib and Cabozantinib are at least partly mediated via MAPK pathway inhibition, it is tempting to speculate that DNA repair status may help guide the selection of systemic therapy for advanced disease. In this study, we did not have access to any HCC patient cohorts with DNA repair-targeting therapy data; thus, we investigated the relationship between DNA repair score and drug response using HCC cell line data (Appendix A). These results indicate that the DNA repair score may help us further understand the role of DNA repair in cancer and establish new drugs that target DNA repair and damage.

This study has some limitations. First, this is a retrospective study utilizing publicly available cohorts, leading to possible selection bias. Further, since we did not have access to a cohort with treatment data, we were unable to investigate the role of the DNA repair score in HCC patient therapy. This study utilized the transcriptome of HCC, which may or may not reflect protein activity. The number of genes used in the score differed because each cohort used different methods of measuring mRNA expression. Despite these limitations, our results were validated by multiple cohorts. Finally, in order to elucidate the underlying molecular mechanisms, in vitro or in vivo experiments are needed; however, the strength of this study comes from our findings in human patients with HCC.

In conclusion, our results suggest that the DNA repair score may be a useful tool to understand the DNA repair pathway and to improve patient outcome.

## 4. Materials and Methods 

### 4.1. Hepatocellular Carcinoma Cohorts and Their Data

mRNA-sequencing data of 358 hepatocellular carcinoma patients in The Cancer Genome Atlas (TCGA) Liver Hepatocellular Carcinoma cohort (TCGA_LIHC, *n* = 358) [6] was downloaded from the Genomic Data Commons Data Portal (GDC). Pathological grade and American Joint Committee on Cancer (AJCC) stage were also obtained from GDC. We used Wurmbach et al. (GSE6764; tumor sample; *n* = 75) [7], Eun et al. (GSE89377; *n* = 107) [48], Brandon et al. (GSE56545; *n* = 42) [49], and Grinchuk et al. (GSE76427; *n* = 167) [8] to investigate the association between the DNA repair pathway scores and HCC patients’ clinico-pathological characteristics and outcomes from the Gene Expression Omnibus (GEO) repository. Pathological classification of the samples in GSE6764 followed the guidelines of the International Working Party [50]. Four pathological HCC stages were defined: (i) very early HCC (8 cases), which included well-differentiated tumors ≤2 cm in diameter with no vascular invasion/satellites (size range: 8–20 mm); (ii) early HCC (10 cases), which included tumors measuring <2 cm with microscopic vascular invasion/satellites; well- to moderately differentiated tumors measuring 2–5 cm without vascular invasion/satellites; or 2–3 well-differentiated nodules measuring <3 cm (size range: 3–45 mm); (iii) advanced HCC (7 cases), which included poorly differentiated tumors measuring >2 cm with microvascular invasion/satellites or tumors measuring >5 cm; and (iv) very advanced HCC (10 cases), which included tumors with macrovascular invasion or diffuse liver involvement. All genomic analysis used normalized transcriptomic data, which were log_2_ transformed. The average value was used for genes with multiple probes. Given that the TCGA and all GEO cohorts used in this study are de-identified in the public domain, approval from the Institutional Review Board was waived.

### 4.2. DNA Repair Scoring Method

The DNA repair score was measured as the gene set variation analysis (GSVA) score for the “Hallmark DNA repair” gene set of the Molecular Signatures Database (MSigDB) Hallmark gene set collections [23] using the GSVA algorithm [51] in the Bioconductor package (version 3.10), as we previously reported [4,5,9,10,11].

### 4.3. Gene Set Expression Analysis

The gene set enrichment analysis (GSEA) algorithm [52], which is publicly available software (GSEA version 4.0.3), with the Hallmark gene sets was used in the study. Statistical significance was determined to a false discovery rate (FDR) of 0.25, as recommended by the GSEA software.

### 4.4. Cell Composition Fraction Estimation

We utilized the xCell algorithm [53] to calculate the fraction of immune cells in the tumor microenvironment through transcriptomic data. The xCell data was obtained through the xCell website (https://xcell.ucsf.edu/), as we previously reported [54,55,56,57,58].

### 4.5. Statistical Analysis

The cytolytic activity score (CYT) was calculated using gene expression of granzyme A (*GZMA*) and perforin (*PRF1*), as previously reported by Rooney et. al. [59]. Microsatellite instability (MSI) data in the TCGA cohort was obtained from GDC. Other score values of the samples in the TCGA cohort, including the intratumor heterogeneity, single-nucleotide variant (SNV) neoantigens, indel neoantigens, silent mutation, non-silent mutation, leukocyte fraction, lymphocyte infiltration, and IFN-γ response score, were calculated and published by Thorsson et al. [14]. The median value of the DNA repair pathway score within cohorts was used to divide the data into low and high DNA repair score groups. Statistical significance for comparison analysis between groups was determined to a *p*-value less than 0.05 by Kruskal–Wallis test, Mann–Whitney U test, and two-tail Fisher’s exact tests. Tukey-type boxplots showed the median and interquartile level values. We utilized R software (version 4.0.1, R Project for Statistical Computing) and Microsoft Excel (version 16 for Windows) for all data analysis and data plotting.

## 5. Conclusions

DNA repair score may be a useful tool to understand the DNA repair pathway and to improve patient outcome.

## Figures and Tables

**Figure 1 cancers-13-00323-f001:**
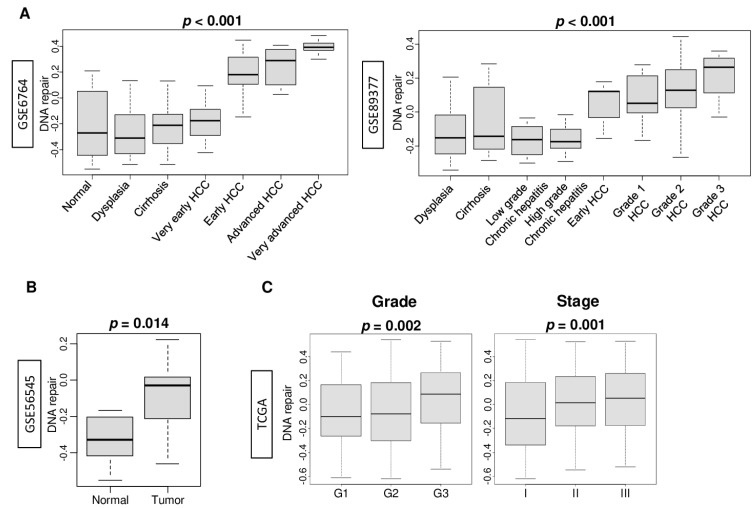
Association between DNA repair and hepatocarcinogenesis. (**A**) Boxplots of the comparison of the DNA repair score by multistep hepatocarcinogenesis, including normal liver tissue (*n* = 8), dysplasia (*n* = 17), cirrhosis (*n* = 13), very early hepatocellular carcinoma (HCC) (*n* = 8), early HCC (*n* = 10), advanced HCC (*n* = 7), and very advanced HCC (*n* = 10) defined by the GSE6764 (*n* = 75) cohort; and dysplasia (*n* = 35), cirrhosis (*n* = 12), low-grade (*n* = 8) and high-grade (*n* = 12) chronic hepatitis, early HCC (*n* = 5), and grades 1–3 (*n* = 9, 12, and 14, respectively) of HCC defined by the GSE98377 (*n* = 107) cohort. (**B**) Boxplots of the comparison of the DNA repair score between normal tissue (*n* = 12) and primary tumor (*n* = 12) in the GSE56545 (*n* = 42) cohort. (**C**) Boxplots of the comparison of the DNA repair score by pathological grade G1–3 (*n* = 53, 168, and 121, respectively) and American Joint Committee on Cancer (AJCC) stage I–III (*n* = 166, 81, and 84, respectively) in the TCGA cohort. The *p*-value was calculated using a Kruskal–Wallis test or Mann–Whitney *U* test.

**Figure 2 cancers-13-00323-f002:**
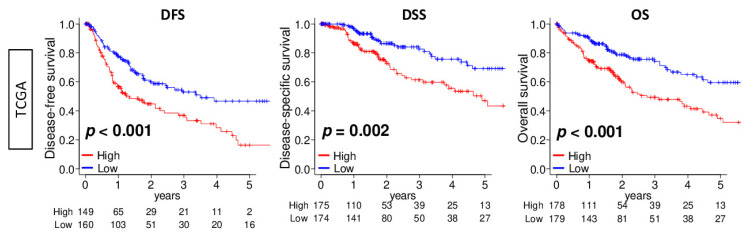
Association between the DNA repair score and HCC patient survival in the TCGA cohort. Kaplan–Meier survival curves comparing the low and high DNA repair score HCC for disease-free survival (DFS), disease-specific survival (DSS), and overall survival (OS) in the TCGA cohort (*n* = 358). We divided the cohort into low and high DNA repair score groups using the median value as the cut-off. The *p*-value was calculated using a log rank test.

**Figure 3 cancers-13-00323-f003:**
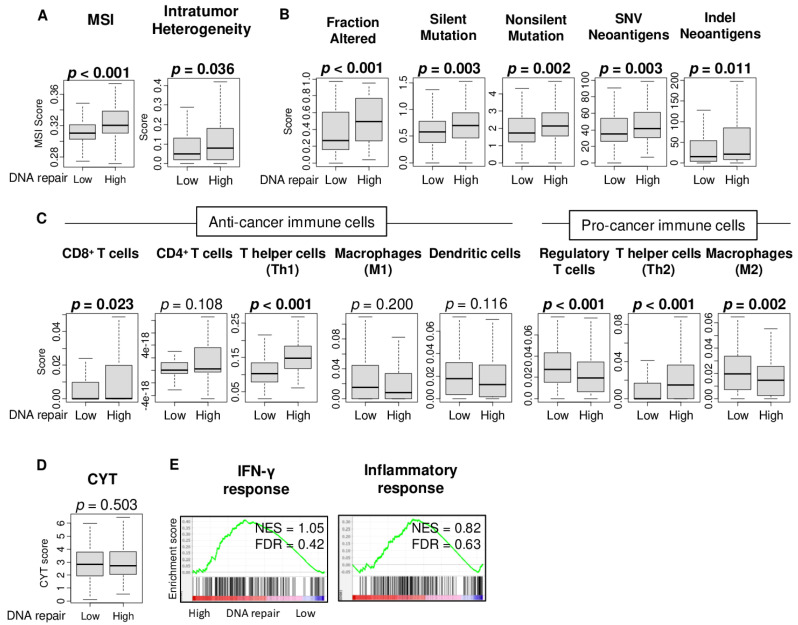
Association between the DNA repair score and microsatellite instability, intratumor heterogeneity, mutation, and infiltrating immune cells in the TCGA cohort. Boxplots of (**A**) the microsatellite instability (MSI) and intratumor heterogeneity score; (**B**) mutation-related score, including the fraction altered, single-nucleotide variants (SNV) and indel neoantigens, and silent and non-silent mutation score. Boxplots of (**C**) the anti-cancer immune cells, including CD8^+^ T cells, CD4^+^ T cells, T helper type 1 (Th1) cells, M1 macrophages and dendritic cells, and pro-cancer immune cells, including regulatory T cells, T helper type 2 (Th2) cells, and M2 macrophages; and (**D**) the cytolytic activity score (CYT) between the low and high DNA repair score groups. The *p*-value was calculated using the Mann–Whitney U test. (**E**) Enrichment plots along with the normalized enrichment score (NES) and false discovery rate (FDR) for the IFN-γ response and inflammatory response gene set of the Hallmark gene sets. Bold format: significant *p* values.

**Figure 4 cancers-13-00323-f004:**
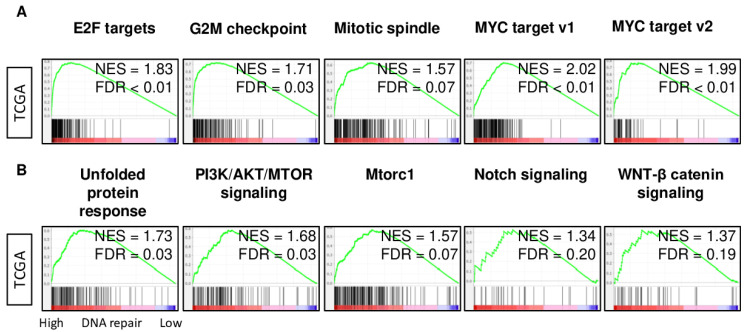
Gene set enrichment analysis (GSEA) of the Hallmark gene sets by high vs. low DNA repair score of HCC in the TCGA cohort. Enrichment plots along with the normalized enrichment score (NES) and false discovery rate (FDR) for (**A**) the proliferation-related gene sets, and (**B**) other gene sets of Hallmark gene sets. An FDR of 0.25 was used to deem statistical significance, as recommended by the GSEA software.

**Figure 5 cancers-13-00323-f005:**
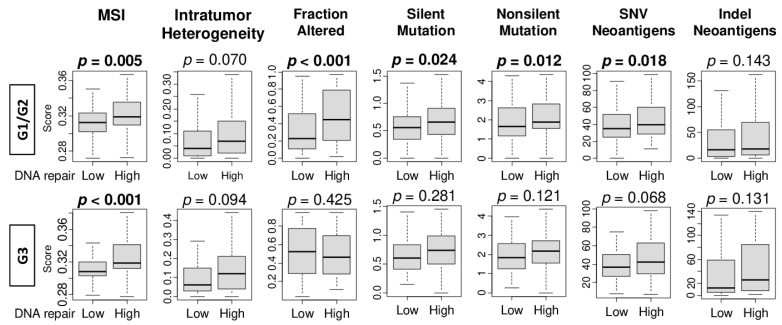
Association of the DNA repair score with microsatellite instability (MSI), intratumor heterogeneity, as well as mutation load by pathological grade 1 and grade 2 (G1/G2) and grade 3 (G3) in the TCGA cohort. Boxplots demonstrate either MSI, intratumor heterogeneity, and mutation-related scores, including the fraction altered, silent and non-silent mutation, single-nucleotide variants (SNV) and indel neoantigens, by low or high DNA repair score in grade 1 and 2 (G1/G2) and grade 3 (G3). The *p*-value was calculated using the Mann–Whitney *U* test. Bold format: significant *p* values.

**Figure 6 cancers-13-00323-f006:**
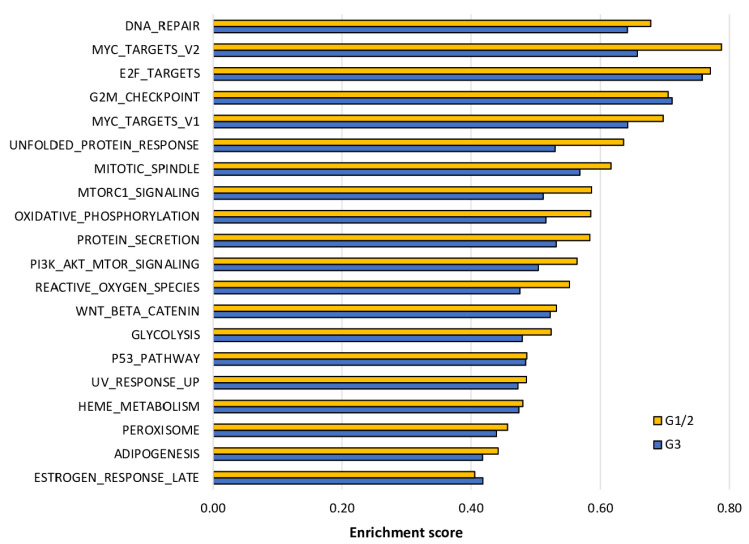
Gene set enrichment analysis (GSEA) of the Hallmark gene sets with significant enrichment due to DNA repair high HCC. Gene sets were listed by the order of high to low enrichment score for the Hallmark gene sets that were statistically significant (false discovery rate (FDR) < 0.25, as recommended by the GSEA software) in grade 1/2. Yellow bars represent the enrichment score of grade 1/2 and the blue bars that of grade 3.

**Figure 7 cancers-13-00323-f007:**
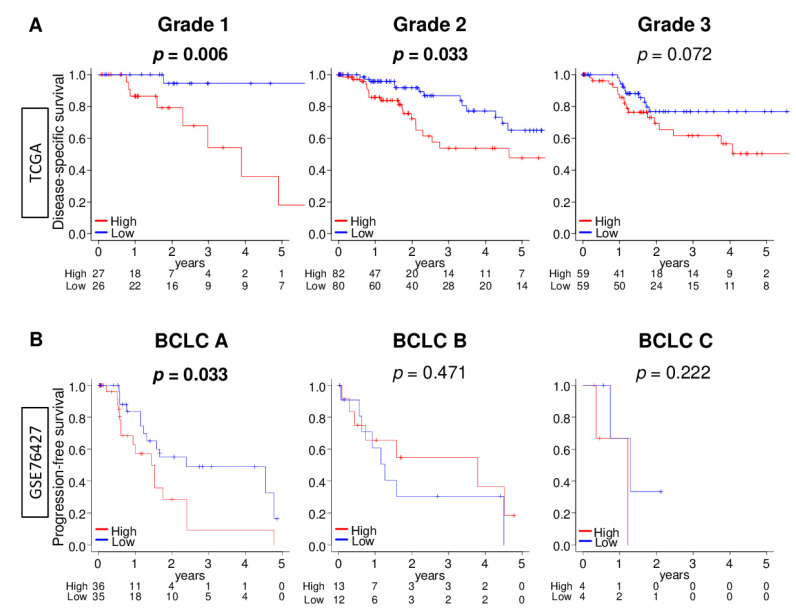
Association of the DNA repair score with patient survival in early and advanced HCC in the TCGA and GSE76427 cohorts. Kaplan–Meier survival curves comparing the low (blue line) and high (red line) DNA repair score tumors for (**A**) disease-specific survival (DSS) in pathological grade 1, grade 2, and grade 3 groups in the TCGA (*n* = 358) cohort; and (**B**) progression-free survival (PFS) in the Barcelona Clinic Liver Cancer (BCLC) stage A, B, and C groups in the GSE76427 (*n* = 167) cohort. We divided the cohort into low and high DNA repair score groups using the median value as the cut-off. The *p*-value was calculated using a log rank test. Bold format: significant *p* values.

## Data Availability

All data were from previous studies.

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
