# Peer review of "Enhanced DNA Repair Pathway is Associated with Cell Proliferation and Worse Survival in Hepatocellular Carcinoma (HCC)"

_cancers, 2021, doi:10.3390/cancers13020323_

Round 1

Reviewer 1 Report

This is an intriguing, well written paper on a topic that could be interesting to a broad audience. However, there are several changes that are necessary in order for the conclusions to be supported by the results.

Comments:

  1. The list of genes included in the Hallmark DNA repair is interesting- it misses known DNA repair proteins, like BRCA1 and BRCA2 (arguably the most famous DNA repair genes associated with cancers) and includes genes that are, to my knowledge, not associated with DNA repair (Eukaryotic Initiation Factor and GPX4 for example). Given that the authors base their entire manuscript on this signature, this warrants some response/changes.
  2. Line 72: “the heterogeneity and immunity of the DNA repair pathway…” what do the authors mean by immunity of the pathway?
  3. Lines 86-88: how large are the unpublished cohorts?
  4. Lines 98-100: how were the datasets separated based on grades or stages? Was it based on IHC type stains? How was this done for the unpublished datasets?
  5. Figure 1: the authors should indicate how many samples were in each stage/grade/classification.
  6. Lines 137-138: I don’t understand the authors’ hypothesis: if DNA repair is high- given that repair is often mutagenic- I would expect higher rates of neoantigens (and this is indeed what the authors do observe).
  7. How were the MSI and tumor heterogeneity scores calculated?
  8. Lines 144-147: This part is very confusing – the authors should clarify what they mean to convey. Why do they expect the Th2 cells to be increased? And what does the fact that those cells are actually down mean for their hypothesis and for the point they are trying to make?
  9. Lines 155-157: I am not sure if I agree with the authors’ conclusion that they see no differences in immune response. The Th1 cells are definitely increased in DNA repair high and all the pro-cancer immune cells are also changing.
  10. Figure 4: did the authors observe MAPK activity also changing in the GSEA analysis?
  11. Line 194: I think the authors’ need to state their hypothesis more clearly. Given that DNA repair is associated with higher grades/more advanced HCC (figure 1) – how do they hypothesize that it is more important in early-stage HCC?
  12. Figure 5: the silent mutations and non-silent mutation graphs between G1/G2 and G3 look very similar – only the “low” samples change between them- not the “high”. In fact, the medians seem even more significant in G3 versus G1/g2. It would be helpful to have the numbers included in each of these analyses.
  13. Figure 6: It is hard to compare the yellow and blue graphs- can this analysis be redone with more comparative statistics?
  14. Figure 7: Is the difference between dna high and low in grade 2 and 3 less significantly difference simply because the disease has progressed further?
  15. The authors note that the IRB approval was waived TCGA and GEO cohorts. Does this also apply to the unpublished cohorts the authors use?

Reviewer 2 Report

The manuscript by Oshi et al.,  investigated the potential role of DNA repair as biomarker in HCC. They analyzed the expression of genes belonging to DNA repair in different cohorts of HCC patients.

This is a retrospective study retrieving data from different cohorts likely to have different patients characteristics and genes analysed.

Although the idea is of potential interest, although not new, I would like to see data on the real response to treatment in HCC,  that is probably the relevant link with DNA Repair status.

I understand that the authors ended the manuscript with “some” limitations. In my opinion these limitations are exactly those which the decrease the potential interest for these data.

It is not surprising that patients with high DNA repair are those with worse prognosis and more aggressive tumors. This is known in several tumors. What is really lacking here is the validation in a properly studied cohort of patients with the possibility to understand the mechanisms behind.

As it this the manuscript would simply add another “potential” biomarker which needs to be further validated. I suspect we need more

Reviewer 3 Report

General Comments:

Manuscript describes the relationship between the enhancement of DNA repair and the cancer aggressiveness, tumor immune microenvironment, and patient survivals, which were examined and validated the HCC with the clinical relevance. Author examined 5 different cohorts, total of 749 HCC patients for their analysis. They hypothesized that cells with excessive DNA damage would not be able to survive, the DNA repair pathway would be enhanced in stepwise carcinogenic process. The analysis procedures were well planned and its interpretation were convincing. However, it is regrettable that the results are based only on bioinformatics/statistics, which gives impression of "speculation" and limited to "nearly" translational research.

For example, authors discussed as L371, "We speculate that tumor immune microenvironment of HCC may not depend on tumor mutational burden and resulting neoantigen load that resulted in lack of association between DNA repair and ~".

I believe that if the author do not have mechanistic evidence to prove the bioinformatic results, they should cite of others to support this conclusion.

This is the same with the speculation that they made on L382.

Mostly, author should emphasize the possible mechanistic difference behind the DNA repair score level defined in the manuscript that would be significant in predicting the prognosis in HCC. Author should strengthen the discussion section to have the paper more convincing.  

Besides, still, results are novel, paper is well written, and the the topic could be interesting to a broad audience in cancer related science. 

Minor Comments:

Abstract: Sentences of L46 "DNA repair high HCC was associated with worse survival, ~", and L50 "DNA repair high HCC was associated with elevated mutation-related score, enriched proliferation- ~" seems to be same indication or just a duplicate. Need to be clarified. 

Round 2

Reviewer 1 Report

The authors have addressed most of my initial concerns, and have clarified their results accordingly. 

Reviewer 2 Report

The authors have their own idea that is of course valid.

Since my opinion has been requested on the manuscript i do not see any improvement based on the comments i made on previous version

My decision remains as it was
